# A Systematic Review of Diagnostic and Prognostic Biomarkers for Head and Neck Cancer of Unknown Primary: An Unmet Clinical Need

**DOI:** 10.3390/diagnostics13081492

**Published:** 2023-04-20

**Authors:** Daria Maria Filippini, Elisabetta Broseghini, Francesca Carosi, Davide Dal Molin, Mattia Riefolo, Laura Fabbri, Andi Abeshi, Ignacio Javier Fernandez, Manuela Ferracin

**Affiliations:** 1Division of Medical Oncology, IRCCS Azienda Ospedaliero-Universitaria di Bologna, 40138 Bologna, Italy; 2Department of Medical and Surgical Sciences (DIMEC), Alma Mater Studiorum, Università di Bologna, 40138 Bologna, Italy; 3Department of Otorhinolaryngology—Head and Neck Surgery, IRCCS Azienda Ospedaliero-Universitaria di Bologna, Policlinico S. Orsola-Malpighi, 40138 Bologna, Italy; 4Pathology Unit, IRCCS Azienda Ospedaliero-Universitaria di Bologna, 40138 Bologna, Italy

**Keywords:** head and neck, cancer of unknown primary, molecular biomarkers

## Abstract

Head and neck cancer of unknown primary (HNCUP) is defined as cervical lymph node metastases without a detectable primary tumor. The management of these patients presents a challenge to clinicians since guidelines in the diagnosis and treatment of HNCUP remain controversial. An accurate diagnostic workup is fundamental for the search for the hidden primary tumor to allow the best adequate treatment strategy. The purpose of this systematic review is to present the currently available data about the diagnostic and prognostic molecular biomarkers for HNCUP. Systematic research in an electronic database was conducted using the Preferred Reporting Items for Systematic Reviews and Meta-Analyses (PRISMA) protocol and identified 704 articles, of which 23 studies were selected and included in the analysis. Fourteen studies investigated HNCUP diagnostic biomarkers and focused on the human papilloma virus (HPV) and the Epstein–Barr virus (EBV) due to the strong associations with oropharyngeal cancer and nasopharyngeal cancer, respectively. HPV status was shown to possess prognostic value, correlating with longer disease-free survival and overall survival. HPV and EBV are the only available HNCUP biomarkers, and they are already used in clinical practice. A better characterization of the molecular profiling and the development of tissue-of-origin classifiers are necessary to improve the diagnosis, staging, and therapeutic management of patients with HNCUP.

## 1. Introduction

Cancers of unknown primary site (CUPs) are rare malignancies defined by the identification of metastatic lesions whose primary site cannot be identified after accurate clinical examination, a surgical approach with biopsy sampling, and imaging techniques. CUPs represent globally about 2–5% of cancers [1]. Within the heterogeneous group of CUPs, the subset of squamous cell carcinoma of unknown primary site in the head and neck (HNCUP) has gained autonomy for its peculiar diagnostic and treatment modalities. HNCUP constitutes 3–5% of all head and neck (H&N) malignancies; however, it accounts for 53–77% of tumors presenting as isolated cervical lymph node metastasis [2]. The risk factors are similar to the ones involved in known primary site H&N tumors, including tobacco smoking and alcohol consumption that act in a synergistic way, male sex, age >40 years, and HPV infection [3].

The HNCUP diagnosis is made by exclusion, as it is represented in Figure 1. The initial diagnostic workup consists of a thorough collection of anamnestic data, such as personal and familiar history of malignancies and the presence of risk factors (e.g., immunosuppressive factors), the clinical symptoms at the onset, diagnostic imaging, and complete H&N physical examination [4,5,6]. The age of onset of H&N cancer peaks at 60–70 years, and males are more affected than females. A cervical mass is usually the only onset symptom of HNCUP. Other associated symptoms could be voice changes, sore throat, hearing loss, ear pain, nasal obstruction, dysphagia, and weight loss. Based on the lymphatic drainage patterns, the metastasis lymph node level can orient clinicians on the most suspicious H&N subsite of origin. Oropharyngeal cancers metastasize more frequently at II and III neck levels. Metastatic lesions involving supraclavicular lymph nodes at IV and VB levels are most often caused by thyroid, lung, or breast carcinomas or neoplasia originated from gastrointestinal, urogenital, and gynecological tracts [3,4,7]. The most common tumor origin sites for each cervical lymph node level are reported in Figure 2.

Contrast-enhanced computed tomography (CT) or magnetic resonance imaging (MRI) of the H&N district should be the first-line imaging test for the workup of a metastatic cervical lymph node followed by positron emission tomography-CT (PET-CT) [2,3,4,5,7,8]. Moreover, patients should undergo a complete upper aerodigestive tract evaluation of the mucosal sites at risk (oral cavity, nasopharynx, oropharynx, hypopharynx, and larynx), including directed biopsy of any suspicious areas. Recent development of surgical options transoral robotic surgery (TORS) and tongue base mucosectomy have added additional therapeutic possibilities [3,9,10]. In the absence of primary tumor detection, immunohistochemistry (IHC) on the biopsy from a cervical lymph node mass can provide vital information for the identification of the primary tumor. First of all, the establishment of the main category of malignancy is needed: sarcoma, lymphoma, melanoma, and carcinoma. By definition, CUPs fall in the last category. Once the epithelial origin is established, it is necessary to identify one out of the four main entities: adenocarcinoma, squamous cell carcinoma, neuroendocrine neoplasm, and poorly differentiated carcinoma.

In the H&N region, useful immunohistochemical analyses are very few. In non-neuroendocrine carcinomas, the HPV status is highly important. In fact, IHC detection of p16, in situ hybridization (ISH), or polymerase chain reaction (PCR) for HPV are implicated in the diagnostic process of HNCUP. Tumor markers allow the differentiation of the broad cancer types, namely, carcinoma, melanoma, lymphoma, and sarcoma, and different subtypes [8,11,12]. It has been largely demonstrated that p16/HPV DNA expression is strongly associated with oropharyngeal origin, a younger age, a better differentiation grade, lower T, a higher risk of lymph nodal extension, a lower probability of infiltration, and a better outcome. ISH detection of the Epstein–Barr encoding region (EBER) could also be helpful in the diagnosis of nasopharyngeal carcinoma (NPC) [13,14,15].

The main goal in clinical management is to eradicate the pathological cervical lymph nodes as well as the hidden primary tumor. A multidisciplinary discussion is mandatory to plan the adequate therapeutic strategy. Since no prospective randomized studies are available for HNCUP patients, no clear statement about the therapeutic strategies is being made [16]. Primary surgery with neck I–IV dissection is usually performed with the purpose of both regional control and prognostic assessment to guide adjuvant therapies. Selective neck dissection (SND) or modified radical neck dissection (MRND) surgery may be a sufficient treatment for N1 necks without extra capsular extension (ECE), but in all other cases, surgery needs to be supplemented by adjuvant radiotherapy and/or chemotherapy [17]. In fact, HNCUP treatment differs depending on the volume of the neck disease. Unilateral or bilateral small-volume neck disease without extracapsular extension (ECE) should undergo surgery or radiotherapy [2]. For bilateral large-volume neck disease, there are different options: surgery followed by radiotherapy, surgery followed by chemoradiation therapy (CRT), definitive CRT, or a range of radiotherapy. Concurrent definitive CRT can be provided to hit the anatomic mucosal regions at risk of hosting the unknown primary tumor as well as the regional lymph nodes (both gross nodal disease and neck nodes which could contain microscopic disease).

RT target volumes used to include the irradiation of bilateral neck and mucosal sites of the entire pharyngeal axes, in order to treat possible occult disease. To date, advances in the field of behavior and biochemistry of head and neck tumors have allowed more risk-tailored approaches with the use of intensity-modulated radiotherapy (IMRT). This allows for a reduction of the clinical target volumes (CTVs) that equates to reduced treatment-related morbidity. Moreover, the expression of p16/HPV-DNA in HNCUP, considering the strong association between HPV positivity and oropharyngeal SCC [18], could justify elective mucosal CTV to the oropharynx, while in EBV-positive HNCUP, in patients with a clinical behavior highly suggestive of an occult nasopharyngeal primitivity, the elective mucosal CTV could be confined to the nasopharynx [19]. In the setting of surgically managed HNCUP, adjuvant RT is often performed, but there are no dedicated randomized clinical trials, and the indication is based on the literature describing the behavior of squamous cell carcinoma arising from known mucosal sites [2]. Patients with metastatic disease, although 10% of patients present this condition at the diagnosis [20], or locoregional recurrence not susceptible to local treatments (surgery and RT) can be treated with a first-line therapy with a combination of platin and 5-FU plus pembrolizumab or alternatively chemotherapy plus cetuximab in the case of HNCUP without the expression of PD-L1 [21].

To date, the diagnosis and treatment of HNCUP remain a challenge. An accurate and meticulous diagnostic workup is fundamental for the search of the hidden primary tumor to allow the best adequate treatment strategy and improve the prognosis. Here a systematic review of the literature is performed to investigate the diagnostic and prognostic role of candidate molecular biomarkers in HNCUP.

## 2. Materials and Methods

### 2.1. Search Strategy

The systematic review was performed in accordance with Preferred Reporting Items for Systematic Review and Meta-Analysis (PRISMA) guidelines [22]. The protocol was registered on PROSPERO (ID: CRD42023405038). We conducted systematic research in PubMed and Web of Science using different combinations of key terms and medical subject headings [MeSH Terms]. An example of the Web of Science search strategy is (ALL = (unknown primary) OR ALL = (occult primary)) AND (ALL = (head and neck) OR ALL = (laryn*) OR ALL = (pharyn*) OR ALL = (oral) OR ALL = (salivary*)) AND (ALL=(rna) OR ALL = (biomarker)). The search strategy in PubMed is summarized with the following search string: (unknown primary) AND ((head and neck) OR (laryn*) OR (pharyn*) OR (oral) OR (salivary*)) AND ((rna) OR (biomarker)).

The reference lists from the screened articles and the review articles were also searched. The cross-references from selected studies were further searched for additional articles. The articles identified through this search were screened and evaluated using identical study selection criteria. The titles and abstracts were independently screened for relevance by seven authors of this study (I.J.F., E.B., D.M.F., F.C., L.F., A.A., and D.D.M.), while disagreements were resolved through discussions with the other authors (I.J.F., E.B., D.M.F., and F.C.).

The electronic database search was conducted in December 2022 without publication year restrictions.

### 2.2. Inclusion and Exclusion Criteria

Studies fulfilling the following criteria were included: original research studies in which the expression of molecular biomarkers in HNCUP tissue or blood was investigated. Studies not written in English or non-original research articles, such as review articles, conference proceedings, editorials, and book chapters, were excluded. Studies describing H&N without focusing on HNCUP patients, studies without HNCUP patients, studies without diagnostic or prognostic molecular biomarkers, and studies without statistically significant data were excluded. The titles and abstracts were independently screened for relevance by six authors of this study (E.B., D.M.F., F.C., L.F., A.A., and D.D.M.), while disagreements were resolved through discussions with the other authors (E.B., D.M.F., F.C., I.J.F., and M.R.). All the selected studies were included in the statistical analysis of the clinical parameters.

### 2.3. Data Extraction

Five independent authors (E.B., F.C., L.F., A.A., and D.D.M.) retrieved information from the eligible articles following the inclusion and exclusion criteria, and the data were collected on a standardized data sheet that included the first author name and publication year, the country where the study took place, the type of study, the number of patients, the biological specimen, the control group, the measurement method, the biomarker type, the biomarker expression, and the outcomes of interest: diagnostic and prognostic. The results of the five independent reviewers were compared, and disagreements on article inclusion and data extraction were resolved by an independent reader.

### 2.4. Data Extraction

The methodologic quality of the included studies was evaluated independently, by five authors, using the Quality Assessment of Diagnostic Accuracy Studies-2 (QUADAS-2) Quality Assessment of Diagnostic Accuracy Studies (QUADAS-2) criteria. If disagreements between the five reviewers occurred, they discussed them together to achieve a consensus or consulted with the sixth reviewer (E.B., D.M.F., F.C., L.F., D.D.M., and I.J.F.).

### 2.5. Data Extraction

SPSS 26.0 software (SPSS, IBM) was used to conduct statistical analysis, while JASP version 0.16.30 was used to obtain the final tables. A descriptive analysis of the variables was carried out. The variables included in the analysis were the biological sample of the extraction of biomarkers, the quantification methods, the type of biomarkers, the treatment response (radiotherapy, chemotherapy, and immunotherapy), and the prognostic parameters (recurrence and disease specific survival). The association of biomarker expression with clinical parameters was displayed in contingency tables. The association between variables was analyzed with chi-square or Fisher’s exact tests as appropriate. The statistical significance was set for *p* < 0.05, and confidence intervals were set at 95%.

## 3. Results

### 3.1. Study Selection

In total, our search identified 704 papers through PubMed (*n* = 474) and Web of Science (*n* = 230) database searching. The first screening removed duplicated (*n* = 150) and in erratum (*n* = 4) papers. Then, we excluded 375 articles for not dealing directly with the investigated issue, 2 records for full-text not being available, 6 for language different to the included one and 20 for not being original papers, such as review, metanalysis and congress abstract. Furthermore, 147 full-text articles were read and then assessed for eligibility. We excluded papers describing HNSCC cases (*n* = 107) not focusing on HNCUP cases. In addition, we excluded papers for not having HNCUP patient or for using patient data from public databases (*n* = 9), for not describing diagnostic or prognostic biomarkers for HNCUP (*n* = 4), and finally, for not showing statistically significant data (*n* = 4). Finally, the review was performed on a total of 23 studies [23,24,25,26,27,28,29,30,31,32,33,34,35,36,37,38,39,40,41,42,43,44,45]. The flowchart of the systematic search is shown in Figure 3. Among the 23 papers, some of them studied CUPs and proposed biomarkers to discriminate CUPs originated from generic H&N primary tumors, namely, HNCUP, from other primary tumor localizations (*n* = 5), while others are focused on HNCUP and aimed to find the specific primary tumor localization among the H&N tumor groups (*n* = 18).

### 3.2. Characteristics of the Selected Studies

The main features of the 23 selected studies are shown in Table 1.

### 3.3. HNCUP Biomarkers

In this systematic review, we selected papers that proposed molecular biomarkers able to discriminate HNCUP from other CUP and able to diagnose the original primary tumor among H&N tumors. Our research identified few biomarkers, and most of the papers described the same biomarkers. Most of the papers analyzed more than one biomarker. Most papers studied biomarkers derived from the analysis of the histologic tumor tissue (*n* = 19), one analyzed cytologic tumor tissue, and two analyzed the peripheral blood. The molecular biomarkers included DNA (*n* = 9), RNA molecules (*n* = 7), and proteins (*n* = 20). Depending on the type of biomarker, different techniques have been used: next generation sequencing (NGS), microarray, polymerase chain reaction (PCR), and in situ hybridization (ISH) or fluorescent in situ hybridization (FISH) were used for DNA and RNA detection, while immunohistochemical staining (IHC) and Western blot were used for protein detection. Circulant proteins were examined through antibodies (Table 2).

#### 3.3.1. Biomarkers Able to Identify H&N Primary Tumor Origin

Two papers studied CUP cases and tried to find some biomarkers able to discriminate those CUPs derived from an H&N primary tumor to others of different primary tumor localizations. A 92-gene assay developed by Raghav et al. was able to identify the tumor types that are sensitive to ICIs in patients with CUPs, including putative H&N primaries (*n* = 1183) [26]. Another assay based on gene expression quantification was proposed by Sun et al.: the 90-gene expression assay identified the primary tumor of CUP patients, and it correctly distinguished the tumor type in 94.4% of specimens, including 31 H&N patients [29].

Three other papers focus on H&N tumors with known or unknown origins. In 45 H&N patients, with known (*n* = 43) and unknown primary (*n* = 2), the levels of endogenous ceramides were measured by high-pressure liquid chromatography/mass spectrometry (LC/MS). Compared with normal tissues, the levels of C(16)-, C(24)-, and C(24:1)-ceramides were significantly elevated in H&N tumor tissues compared with their normal tissues, while C(18)-ceramide levels were significantly decreased. A higher level of C(18)-ceramide was significantly associated with higher incidences of lympho-vascular invasion, pathologic nodal metastasis, and a worse prognosis [27].

The sera of 138 patients with known H&N primary locations, including the larynx (*n* = 50), hypopharynx (*n* = 34), oropharynx (*n* = 38), oral cavity (*n* = 11), and skin (*n* = 5), and the sera of five HNCUP patients were investigated for the presence of p53 antibodies. After the analysis, the authors proposed the detection of serum p53 antibodies in H&N patients as a biological marker to identify patients that are at high risk of developing recurrences and/or second primary tumors and at high risk of undergoing therapy failure [28]. Prognostic information about these molecular biomarkers is reported in Table 3.

Comparing HNCUPs with the metastasis of known H&N primaries, a lower VEGF expression, specifically of VEGF121 and VEGF165 isoforms, at the immunohistochemical and protein level was found in HNCUP, suggesting that HNCUPs could have a low angiogenic phenotype and so might not respond to antiangiogenic therapy [36].

#### 3.3.2. Head and Neck Cancer of Unknown Primary Diagnostic Biomarkers

Fourteen papers have investigated HNCUP cases and proposed human papilloma virus (HPV) infection and Epstein–Barr virus (EBV) infection as diagnostic biomarkers. In fact, these infections are associated with different sites among H&N tumors. Specifically, HPV is associated with oropharyngeal cancer (OPC), while EBV is often associated with NPC. Table 4 reported the number of papers that associated a biomarker with a specific diagnosis in HNCUP patients.

HPV biomarker in HNCUP diagnosis

Eight papers reported the identification of the primary tumor of HNCUP by analyzing HPV. Park et al. enrolled 58 patients with cervical lymph node metastases from HNCUP and tested them for HPV by ISH and for p16 and p53 expression by IHC. The presence of HPV and p16 was associated with OPC location. The results of HPV ISH and p16 IHC inversely correlated with p53 IHC. The OPC primary was found in 20 patients [23]. Five years later, the same groups published another paper where they showed the effectiveness of HPV testing to identify occult primary tumors undetectable by the conventional diagnostic technique based on 18 F-fluorodeoxyglucose (18 F-FDG) positron emission tomography/computed tomography (PET/CT) [25]. The tissues of lymph node metastases of 47 HNCUP patients were analyzed for the presence of HPV DNA through PCR. The PCR-based HPV status was confirmed by p16 IHC. Their data showed that HPV-positive lymph node metastasis suggested OPC as the primary tumor [38].

HPV DNA detection by PCR and p16 evaluation by IHC for an OPC diagnosis was also performed by another group, who found that HPV-positive carcinomas, including HNCUP, appeared to arise from multiple sites in the oropharynx, particularly the tonsils and tongue base [35]. In addition to p16 IHC, another group detected HPV DNA by ISH instead of PCR. In this case as well, the p16-positive lymph node metastases contained significantly more HPV DNA and were most frequently associated with occult primary lesions in the oropharynx [41].

Shan et al. revaluated 20 HPV-positive HNCUPs that were interpreted as negative for OPC at the time of the original diagnosis. Through p16 IHC, they confirmed the oropharyngeal primary origin for two of these patients (10%) [37]. In two case reports, the oropharyngeal primary origin was identified through p16 IHC [30]. In addition to detecting HPV in tissues, by using PCR for HPV DNA and IHC for p16, Schroeder et al. investigated HPV in sera of 46 HNCUP. Specifically, they used antibodies against early HPV proteins, namely, E6 and E7, which belonged to several HPV strains (HPV16/18/31/33/35), and E1 and E2, produced by HPV16 and HPV18 strains. They observed that HPV seropositivity appeared to be a consistent diagnostic biomarker for OPC [39].

EBV biomarkers in HNCUP diagnosis

EBV genomic DNA and EBV RNA from the Epstein–Barr encoding region (EBER) are associated to NPC [25]. Dictor et al. analyzed EBV RNA expression by EBV early ribonucleoprotein 1 (EBER1) ISH in metastatic tissues obtained from 18 nasopharyngeal, 54 oropharyngeal, and 12 HNCUP carcinomas. They observed that all nonkeratinizing NPCs and a single positive case of HNCUP were positive for EBER1, while the other patients were all EBV negative [34]. Using EBV RNA ISH, Nakao et al. examined 36 NPCs, including 30 primary tumors and six metastasized lymph nodes, 13 metastasized lymph nodes of other H&N cancers, and 12 HNCUPs. The majority of NPCs and one HNCUP were EBV positive, while EBV was not detected in any metastasized lymph nodes of other H&N cancers [45].

A different diagnosis of EBV-positive HNCUP was observed in a Chinese case report. The study described an HNCUP patient with EBV positivity, which was defined through EBER ISH, but the examination and biopsy of her nasopharynx were negative. The authors reported a case of an EBV-positive undifferentiated carcinoma with lymphoid stroma in the lacrimal sac, suggesting that the lacrimal sac should be considered as a potential primary site for EBV-positive HNCUP [33].

Other HNCUP diagnostic biomarkers

A case report described an HNCUP patient with the presence of human salivary alfa-amylase mRNA in an H&N tumor detected by ISH. The primary tumor was most likely a heterotopic salivary gland adenocarcinoma (HSGA); however, it was not found macroscopically in the major salivary glands at autopsy [31]. Another case report found an HNCUP positive for cytokeratin-7 (CK7) and GATA binding protein 3 (GATA3) and negative for transcriptional thyroid factor 1 (TTF1) and transformation-related protein 63 (p63). A diagnosis of metastatic epithelial–myoepithelial carcinoma was provided [32].

Head and neck cancer of unknown primary prognostic biomarkers

In addition to being used as a diagnostic biomarker, HPV status can provide useful prognostic indications. Park et al. showed that p16 expression and HPV detection correlated with longer disease-free survival (DFS) and overall survival (OS), while high p53 expression predicted worse prognosis with shorter DFS and OS [23]. Sivars et al. enrolled 19 HNCUP patients and analyzed HPV DNA by PCR and p16 by IHC. To confirm HPV-driven carcinogenesis in HNCUP, they also quantified HPV16 mRNA by PCR, which was present in most HPV16 DNA-positive cases. HPV DNA, alone and in combination with p16 overexpression, was validated as a favorable prognostic factor in HNCUP [40]. A total of 180 HNCUP patients were enrolled in a European multicenter retrospective study and were examined for the presence of HPV DNA and HPV E6*I mRNA by PCR and p16 overexpression by IHC. The study correlated HPV status, defined as positivity for viral mRNA with at least one additional marker (HPV DNA or p16), with prognostic parameters and showed that HPV-driven HNCUP had significantly better overall and progression-free survival rates [44]. Among patients with HNCUP, p16-positive status has been described an independent predictor of DFS [25,42] and for OS [43]. The prognostic HPV biomarker was also described in HNCUP sera. In fact, the HPV detection through antibodies in HNCUP patients showed that HPV-seropositive patients had a better overall and progression-free survival [39].

A different prognostic biomarker was described by Park et al. They observed that retinoblastoma protein (Rb) expression of metastatic lymph nodes represented an independent prognostic indicator in HNCUP patients. Specifically, they examined protein expression by IHC, which was associated with DFS and OS that were shorter in Rb-positive tumors [24].

Prognostic data about molecular biomarkers in HNCUP patients are reported in Table 5. The association between the biomarker (Table 6) or between the biomarker group (Table 7) and the head and neck subsites are reported.

## 4. Discussion

HNCUP is defined as cervical lymph node metastases without a detectable primary tumor. The management of these patients presents a challenge to clinicians since guidelines in the diagnosis and treatment of HNCUP remain controversial [16]. An accurate diagnostic workup is fundamental for the search for the hidden primary tumors, to allow the best adequate treatment strategy. In this systematic review, we aimed to report the current data about the diagnostic and prognostic biomarkers for HNCUP.

Gene expression panels have been tested for the identification of primaries of CUP patients and were able to predict H&N as primary tumor [26,29]. Focusing on H&N tumors, it was discovered that HNCUPs and the metastasis of known H&N primaries present a different angiogenic phenotype; specifically, HNCUPs seem to present a lower angiogenic phenotype suggesting a worse response to antiangiogenic therapy [36].

Most of the papers collected in this systematic review are focused on diagnostic biomarkers that can provide an indication of the HNCUPs’ primary tumor location. The two main groups of biomarkers are related to HPV and EBV testing.

The infectious agent HPV is mainly sexually transmitted and may lead to HPV-associated cancers. In fact, it is an established cancer-promoting factor for cervical cancers and some H&N cancers, especially squamous cell carcinomas of the oropharynx [46]. Even if HPV-positive oropharyngeal cancer cells are often histologically characterized by a non-keratinizing or basaloid morphologic pattern [47], cell morphology alone is lacking to confirm HPV presence. The direct diagnosis of HPV infection can be performed by the detection of HPV DNA or mRNA through PCR and ISH. Since these two techniques do not provide information about the biological activity of the HPV infection, many studies used the immunohistochemical detection of p16 as a marker for active HPV viral oncoproteins, as well. In fact, HPV oncoproteins lead to the inactivation of cell cycle control and a consequent overexpression of p16 [48]. However, p16 positivity is not limited to HPV-positive tumors, and therefore, it is not a perfect surrogate for HPV [49]. The analysis of HPV infection in H&N tumor samples is crucial since it can direct the research of the primary tumor of HNCUP. In our systematic review, we reported the results of 12 papers that diagnosed the primary tumor of HNCUP as oropharynx after HPV infection investigation. Eight papers used the combination of direct HPV detection by DNA or mRNA quantification and indirect HPV detection by p16 IHC, including one paper that also used HPV detection in serum, while four papers used only p16 ISH. All papers that studied the potential prognostic value of HPV infection agreed to indicate a better prognosis, in terms of DFS and OS, in HPV-positive tumors compared with HPV-negative tumors.

Another infectious agent that may lead to H&N tumors is EBV. In fact, latent EBV infection is known to be a promoter of the development of multiple lymphoid malignancies and epithelial cancers, including NPC [50]. NPCs are often diagnosed with local-advanced or distant metastasis since early-stage patients with NPC are usually asymptomatic. The detection of specific EBV DNA and antibodies is important and crucial for the early diagnosis of NPC [51]. Many papers described an association between EBV infection and NPC [25,34,45]. Only one paper reported a diagnosis of EBV-positive undifferentiated carcinoma with lymphoid stroma [33].

In the clinical management of HNCUP, both HPV and EBV detections are already used to identify the primary tumor of the oropharynx and NPC, respectively (Figure 1). In fact, according to the 8th edition of the TNM Classification of Malignant Tumors from the AJCC/UICC, p16 IHC and EBER-ISH represent essential tests for the diagnosis of the tumor type in HNCUP patients [52].

Other biomarkers were described in single studies by a single group, and they need to be validated in larger HNCUP cohorts to be considered as future biomarkers in HNCUP patient management.

The emerging use of liquid biopsies as a source of diagnostic and prognostic biomarkers, including extracellular vesicles, in known tumor types has shown a huge potential for the improvement of patient management [53,54,55], which needs to be translated in future studies and clinical practice of CUP [56] and HNCUP patients, to improve life expectancy and quality of life. In this systematic review, we found and reported only a paper that uses HPV sera protein as a diagnostic and prognostic biomarker in HNCUP [39].

A crucial role in cancer onset and progression, including for H&N tumors, and a potential utility as diagnostic or prognostic biomarkers are displayed by small non-coding RNAs and especially miRNAs [57,58]. From our research and to our knowledge, there is no paper that uses an miRNA expression panel to discriminate the specific H&N location in HNCUP with statistically significant data. However, an miRNA expression profiling panel to determine the primary tumors of CUPs has been already developed by Laprovitera et al. [59]. This serves as the basis for future studies on the miRNA role in HNCUP patients, which could lead to the discovering of new diagnostic and prognostic biomarkers. A better characterization of the molecular profiling and the development of tissue-of-origin classifiers are necessary to improve the diagnosis, staging, and therapeutic management of patients with HNCUP. The era of precision medicine will offer improvements in diagnosis and therapy for HNCUP patients.

## 5. Conclusions

The literature on diagnostic and prognostic biomarkers in HNCUP is limited. Currently, only HPV and EBV are confirmed diagnostic or prognostic biomarkers for HNCUP, useful in the clinical setting. Therefore, more studies focusing on the identification of diagnostic and prognostic molecular biomarkers are required to select those that are clinically relevant and that can be translated into the clinical practice.

## Figures and Tables

**Figure 1 diagnostics-13-01492-f001:**
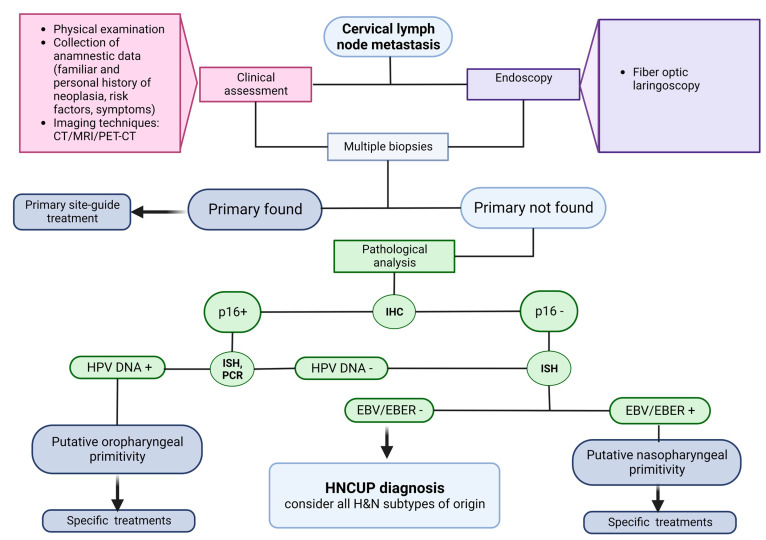
HNCUP diagnostic procedure. Figure created with Biorender.com.

**Figure 2 diagnostics-13-01492-f002:**
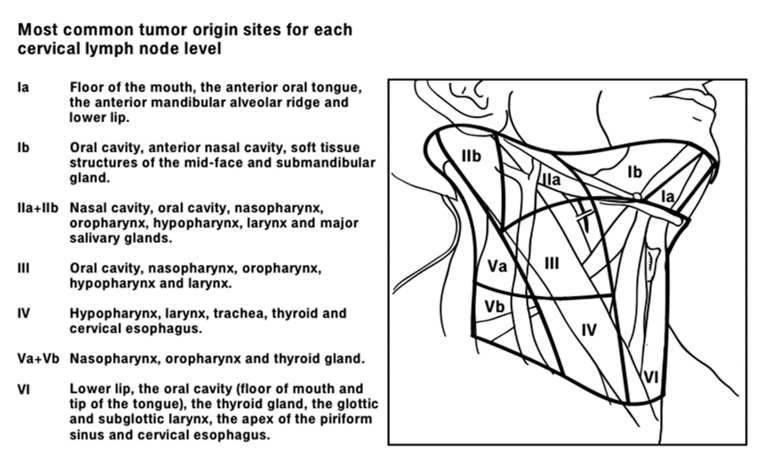
First lymph node levels of lymphatic drainage according to the head and neck tumor site.

**Figure 3 diagnostics-13-01492-f003:**
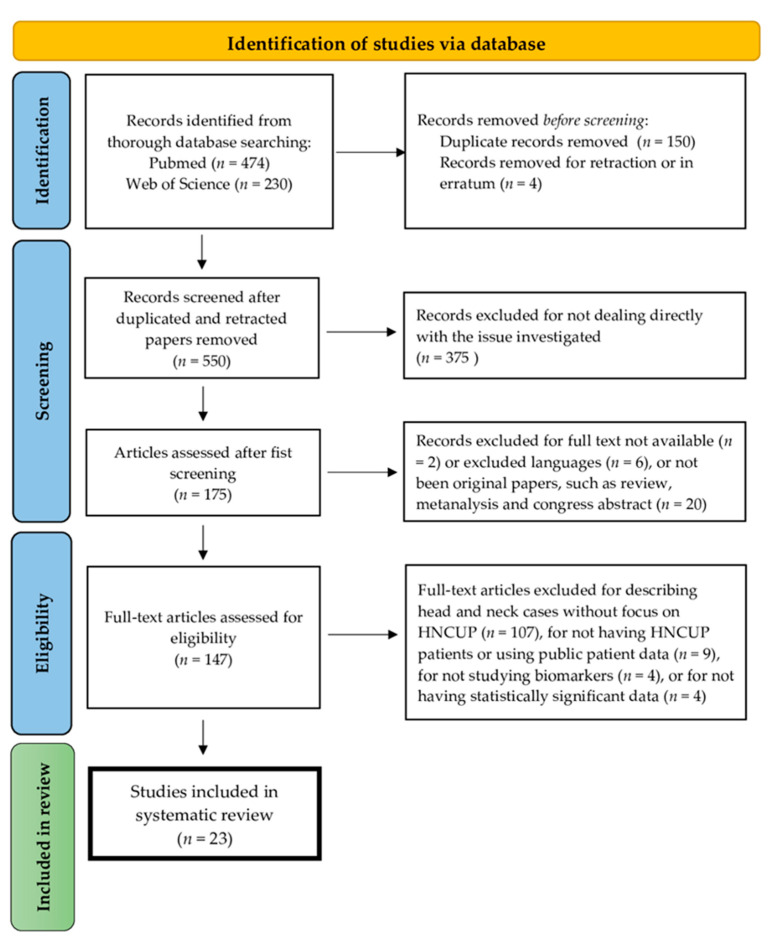
PRISMA flow diagram showing the steps of the systematic review of the literature. Of 704 papers, 23 original papers were selected in this systematic review.

**Table 1 diagnostics-13-01492-t001:** Description of general features of the 23 selected studies in this systematic review.

Case Description		
	Frequency *	Percentage
**General features**		
**Publication Date**		
Before 2000	4	17.4
2000–2010	3	13
2011–2019	12	52.2
After 2020	4	17.4
Total	23	100
**Country**		
Canada	1	4.4
China	2	8.7
Germany	3	13
India	1	4.4
Korea	3	13
Japan	3	13
Sweden	2	8.7
US	7	30.4
Multicenter (Germany, Italy and Spain)	1	4.4
Total	23	100
**Study type**		
Prospective	2	8.7
Retrospective	15	65.2
Case report	4	17.4
Prospective and retrospective	2	8.7
Total	23	100

* Frequency corresponds to number of papers.

**Table 2 diagnostics-13-01492-t002:** Description of biomarkers of the 23 selected studies in this systematic review.

	Frequency *	Percentage
**Biological sample**		
Histologic tumor tissue	19	82.5
Cytologic tumor tissue	1	4.4
Histologic and cytologic tumor tissue	1	4.4
Peripheral blood	2	8.7
Total	23	100
**Quantification methods**		
Next generation sequencing or microarray	1	3
Real time quantitative PCR	9	27.3
Immunohistochemical staining	13	39.4
In situ hybridization (ISH) or fluorescent ISH (FISH)	7	21.2
Western blot or antibodies	3	9.1
Total **	33	100
**Type of molecular biomarkers**		
DNA	9	25
RNA	7	19.4
Protein	20	55.6
Total **	36	100
**Biomarkers list**		
p16	11	33.4
HPV DNA	7	21.2
HPV RNA	1	3
Circulating HPV protein	1	3
EBV DNA/RNA	4	12.2
p53/p53 antibodies	2	6.1
Genetic panel assay	2	6.1
Ceramides (*n =* 4)	1	3
Salivary mRNA	1	3
VEGF	1	3
Rb	1	3
GATA 3	1	3
Total **	33	100

* Frequency corresponds to number of papers. ** Some papers studied more than one biomarker; for this reason, the total is higher than 23 for the type of molecular biomarkers and quantification methods.

**Table 3 diagnostics-13-01492-t003:** Molecular biomarkers provided from CUP studies and associated with prognostic features in H&N patients. ↑= upregulated; ↓= downregulated.

Biomarker	Disease-Free Survival	Overall Survival	RT Response	Chemo Response	Immunotherapy	Number of Paper(s)	Refs.
92-gene assay	-	-	-	-	1 ↑	1	[26]
p53 antibodies	1 ↓		1 ↓	1 ↓		1	[28]
C(18)-ceramide		1 ↓	-	-	-	1	[27]

**Table 4 diagnostics-13-01492-t004:** Molecular biomarkers with diagnostic association in HNCUP.

Biomarker	Sample Size	Diagnostic Association N°	Over-Expressed N° (%)	Down Regulated N° (%)	Refs *
p16	167/391	5/9	5/9 (55.5)	0/9	[23,25,30,37,41]
HPV DNA/RNA	200/495	4/7	4/7 (57.1)	0/7	[23,25,35,38]
EBV DNA/RNA	79/79	4/4	3/4 (75)	1/4 (25)	[25,33,34,45]
P53/p53 antibodies	63/63	2/2	1/2 (50)	1/2 (50)	[23,28]
Genetic panel assay	1214/1214	2/2	-	-	[26,29]
Ceramides	8/8	4/4	3/4 (75)	1/4 (25)	[27]
Salivary mRNA	1/1	1/1	1/1 (100)	0	[31]
VEGF	50/50	1/1	0	1/1 (100)	[36]
Other	1/ 37	2/2	1/2 (50)	1/2 (50)	[24,32]
TOTAL	1783	25/32	18/32	5/32	

N°: number of studies reporting the diagnostic association. Other: pRb and GATA3. Sample size: number of patients with diagnostic association/total number of patients on which the biomarker has been studied. Diagnostic association: N° of papers showing diagnostic association/N° of total analyzed papers dealing with the mentioned biomarker. * Concerning only the N° of papers dealing with diagnostic association of the mentioned biomarker.

**Table 5 diagnostics-13-01492-t005:** Molecular biomarkers with prognostic association in HNCUP.

Biomarker	Sample Size	Prognostic Association	Recurrence N°	OS N°	RT Response	Chemo Response	Refs. *
p16	336/358	Yes 6/8	4 ↑	4 ↑	0	0	[23,25,40,42,43,44]
HPV DNA/RNA	353/495	Yes 4/7	3 ↑	4 ↑	0	0	[23,39,40,44]
EBV DNA/RNA	0/79	No 0/4	0	0	0	0	
P53/p53 antibodies	63/63	Yes 2/2	1 ↓	1 ↓	1 ↓	1 ↓	[23,28]
Genetic panel assay	1183/1214	Yes **(predictive)** 1/2	1 ↑	-	0	1 ↑	[26]
Ceramides	2/8	Yes 1/4	0	1 ↓	0	0	[27]
Salivary mRNA	0/1	No 0/1	0	0	0	0	
VEGF	0/50	No 0/1	0	0	0	0	
Other	1/37	Yes 1/2	1 ↓	1 ↓	0	0	[24]
TOTAL	790	14/31	9	11	1	2	

N°: number of studies reporting the association. Other: pRb and GATA3. ↑: high/positive expression if the biomarkers are associated with a better prognosis. ↓: high/positive expression if the biomarkers are associated with a worse prognosis. Sample size: number of patients with prognostic association/total number of patients on which the biomarker has been studied. Prognostic association: N° of papers showing prognostic association/N° of total analyzed papers dealing with the mentioned biomarker. * Concerning only the N° of papers dealing with prognostic association of the mentioned biomarker.

**Table 6 diagnostics-13-01492-t006:** Association between biomarker and head and neck subsite.

Marker	Sample Size	Oropharynx N (%)	Nasopharynx	Salivary Gland	Head and Neck (Generic)	Two or More Head and Neck Subsites	*p* Value
p16	167/391	5/9 (55.6)	0	0	0	0	**0.031**
HPV	200/495	3/7 (42.8)	0	0	0	1/7 (14.3)	**0.327**
EBV	79/79	0	2/4 (50.0)	0	2/4 (50.0)	0	**<0.001**
p53/p53 antibodies	63/63	1/2 (50.0)	0	0	1/2 (50.0)	0	**0.477**
Genetic panel assay	1214 /1214	0	0	0	2/2 (100)	0	**-**
Ceramides	8/8	0	0	0	4/4 (100)	0	**-**
Salivary mRNA	1/1	0	0	1/1 (100)	0	0	**0.053**
VEGF	50/50	0	0	0	1/1 (100)	0	**-**
other	1/1	0	0	1/2 (50.0)	0	0	**-**

HPV includes HPV DNA, RNA, and circulating HPV proteins; EBV includes EBV DNA and RNA (EBER and EBER1 RNA). Genetic panel assay: 92- and 90-gene panel assay [26,29]. Ceramides: 16, 18, 24, 24:1; VEGF:VEGF 161, and 165. Other: pRb and GATA3. *p* values have not been calculated for generic association with head and neck cancer (no subsite analysis reported). Sample size: number of patients supporting the subsite(s) association/total number of patients on which the biomarker has been studied.

**Table 7 diagnostics-13-01492-t007:** Association between biomarkers group and head and neck subsites.

Biomarker	Head and Neck Subsite	*p* Value	Odds Ratio (95% CI)
HPV-related biomarkers	Oropharynx	0.015	2.71 (0.46–4.97)
EBV-related biomarkers	Nasopharynx	0.012	4.04 (0.74–7.64)
Salivary-gland-related biomarkers	Salivary gland	0.002	5.72 (1.59–9.84)

HPV-related biomarkers: HPV DNA, RNA, circulating HPV proteins, and p16. EBV-related biomarkers: EBV DNA and RNA (EBER and EBER1 RNA). Salivary-gland-related biomarkers: salivary mRNA and GATA3.

## Data Availability

This study did not generate a data archive. All data are already included in the manuscript.

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
