# Peer review of "A Systematic Review of Diagnostic and Prognostic Biomarkers for Head and Neck Cancer of Unknown Primary: An Unmet Clinical Need"

_diagnostics, 2023, doi:10.3390/diagnostics13081492_

Round 1

Reviewer 1 Report

See the file attached. The review may be accepted for publication without revision.

Author Response

THANK YOU VERY MUCH TO REVIEWER 1  FOR YOUR KIND RESPONSE. 

Reviewer 2 Report

This is a thorough review of potential biomarkers of use in defining the origin of squamous carcinomas from an unknown primary in the head and neck region. The impact of this review would be improved by some expansion of the Introduction to show the basis of the controversies about management and how biomarker might aid treatment selection and planning. In places there is repetition which adds unnecessarily to the length of the paper.

Controversies about management stem from the uncertainty of the primary site and whether it is necessary always to include this as part of the treatment plan. The strategy of only treating the neck and leaving the primary site to declare itself risks progression at the primary site (though perhaps the risk of this is lower in the era of PET/CT). However, studies of neck only treatment suggests that the primary site does not always declare itself, opening the possibility that some small primary tumours may spontaneously regress. The other factor in the controversy is that if radiotherapy is given to the neck, in most cases this results in potential primary sites being in low dose regions, and then if a primary site does declare itself it may prove impossible to deliver an effective dose (essentially re-treatment) thereby committing the patient to a surgical approach which may result in an ultimately worse outcome, certainly in terms of swallowing function. It is this that has led to chemoradiotherapy to the neck and potential primary sites being the standard of care in many centres in all but the earlies stages of disease. The results of your review are particularly valuable here. Several decades ago, it was standard practice to include all potential primary sites from nasopharynx down to supraglottis whereas now the treatment volume can be reduced on the basis of P16 and EBV studies. Reducing treatment volume equates with reduced treatment-related morbidity. These points should be brought out in the Introduction.

Recent development of surgical options TORS and tongue base mucosectomy have added additional therapeutic possibilities. There are, strictly speaking, therapeutic options and not diagnostic tools so should perhaps be omitted from Figure 1 and rephrased in lines 76-78.

There is some repetition in 88-89 and 91-92 which say the same thing.

Clinical management (lines 98-121): you say there are “no unique guidelines” in line 100 but go on to cite UK guidelines (reference 15) in line 107.

EBV involvement: references 9, 10 and 13 (line97) all relate to HPV not EBV. References that relate to EBV are required here.

IMRT in HNCUP is “often” performed (line 115). IMRT has been the standard of care for all H&N radiotherapy of this type for at the least the last 10 years.

Line 118: “patients with metastatic disease or locoregional recurrence”. It would be helpful to state (and reference) that less than 10% have metastatic disease at presentation and that following optimal management only around 10% will have locoregional recurrence.

Results:

Your use of tables is good and very helpful to the reader. However, it is not necessary to repeat all the data in the text. Lines 186-196 and lines 207-212 could be significantly truncated. Line 264 could be omitted.

Figure 3 should be labelled PRISMA flow diagram.

Line 237: the meaning of this sentence is unclear.

Tables 4, 5 and 6 presumably all relate to data by number of papers, although that is not always stated in the Table legend. In Table 5 the number of papers does not seem to correspond to the number of references in the far-right column. It would be helpful to know how many patients contributed to this data. Perhaps that might be included in an additional column?

References:

The reference to PRISMA guidelines (line 130; ref 17) needs correcting.

There are various places in the text with numbers in brackets (e.g 32324430 in line 109) which presumably are references. Lines 290, 293, 294 likewise.

Minor points:

Line 50: it is unclear what exactly you mean by anamnestic in this context.

Line 165: data extraction section: “disagreements on search strategy”; surely that should have been resolved before carrying out the search?

Line 287: tongue base rather than tongues.

Author Response

Dear Editor and Dear Reviewer:

We express our deepest gratitude for the constructive comments and suggestions. We really appreciate your time and efforts spent in reviewing our manuscript for possible publication in your prestigious journal. We are more than pleased to accept your critiques and did our best to provide valid and convincing explanations in this letter.

Below is the point-by-point response to the comments and suggestions. The changes are highlighted in yellow color in the manuscript. If you have any further recommendations, please let us know, and we will provide relevant revisions. Thank you.

1)This is a thorough review of potential biomarkers of use in defining the origin of squamous carcinomas from an unknown primary in the head and neck region. The impact of this review would be improved by some expansion of the Introduction to show the basis of the controversies about management and how biomarker might aid treatment selection and planning. In places there is repetition which adds unnecessarily to the length of the paper. Controversies about management stem from the uncertainty of the primary site and whether it is necessary always to include this as part of the treatment plan. The strategy of only treating the neck and leaving the primary site to declare itself risks progression at the primary site (though perhaps the risk of this is lower in the era of PET/CT). However, studies of neck only treatment suggests that the primary site does not always declare itself, opening the possibility that some small primary tumours may spontaneously regress. The other factor in the controversy is that if radiotherapy is given to the neck, in most cases this results in potential primary sites being in low dose regions, and then if a primary site does declare itself it may prove impossible to deliver an effective dose (essentially re-treatment) thereby committing the patient to a surgical approach which may result in an ultimately worse outcome, certainly in terms of swallowing function. It is this that has led to chemoradiotherapy to the neck and potential primary sites being the standard of care in many centres in all but the earlies stages of disease. The results of your review are particularly valuable here. Several decades ago, it was standard practice to include all potential primary sites from nasopharynx down to supraglottis whereas now the treatment volume can be reduced on the basis of P16 and EBV studies. Reducing treatment volume equates with reduced treatment-related morbidity. These points should be brought out in the Introduction.

Response 1) Thank you for this insightful comment. We provided to add in the introduction these important issues.

2) Recent development of surgical options TORS and tongue base mucosectomy have added additional therapeutic possibilities. There are, strictly speaking, therapeutic options and not diagnostic tools so should perhaps be omitted from Figure 1 and rephrased in lines 76-78.

Response 2) We provided to modify the figure as you correctly suggest us omitting “new techniques” and we rephrased in lines 76-78 as follows “Recent development of surgical options transoral robotic surgery (TORS) and tongue base mucosectomy have added additional therapeutic possibilities”.

3) “There is some repetition in 88-89 and 91-92 which say the same thing.

Response 3) Thank you, we truncated the line 91 and 92.

4) Clinical management (lines 98-121): you say there are “no unique guidelines” in line 100 but go on to cite UK guidelines (reference 15) in line 107

Response 4) Thank you very much for your insightful comment, we changed the sentence in line with our initial interpretation, as follows “Since no prospective randomized studies are available for HNCUP patients, no clear statement about the therapeutic strategies is being made”.

5) EBV involvement: references 9, 10 and 13 (line97) all relate to HPV not EBV. References that relate to EBV are required here.

Response 5) We added the correct references related to EBV that are now reported as [13-15].

6) IMRT in HNCUP is “often” performed (line 115). IMRT has been the standard of care for all H&N radiotherapy of this type for at the least the last 10 years.

Response 6) We really appreciate this percipient comment. Our previous interpretation refers only for the adjuvant setting of IMRT. However, we provided to change the sentence to make it better comprehensible. The sentence is now reported in line 129 and modulated as “In the setting of surgically managed HNCUP, adjuvant RT is often performed, but there are no dedicated randomized clinical trials, and the indication is based on literature describing the behavior of squamous cell carcinoma arising from known mucosal sites.”

7) Line 118: “patients with metastatic disease or locoregional recurrence”. It would be helpful to state (and reference) that less than 10% have metastatic disease at presentation and that following optimal management only around 10% will have locoregional recurrence.

Response 7) We changed the sentence adding what you correctly asked us (“Patients with metastatic disease, although a 10% of patients present this condition at diagnosis…”) and we added the reference [20].

Results:

8) Your use of tables is good and very helpful to the reader. However, it is not necessary to repeat all the data in the text. Lines 186-196 and lines 207-212 could be significantly truncated. Line 264 could be omitted.

Response 8) We appreciated this useful suggestion. We provided to truncate these lines.

9) Figure 3 should be labelled PRISMA flow diagram.

Response 9) The figure 3 was labelled as PRISMA flow diagram.

10) Line 237: the meaning of this sentence is unclear.

Response 10) Thank you, we provided to modify the sentence as follows “A 92-gene assay developed by Raghav et al. was able to identify tumor types that are sensitive to ICIs in patients with CUP, including putative H&N primaries (n=1183)”.

11) Tables 4, 5 and 6 presumably all relate to data by number of papers, although that is not always stated in the Table legend. In Table 5 the number of papers does not seem to correspond to the number of references in the far-right column. It would be helpful to know how many patients contributed to this data. Perhaps that might be included in an additional column?

Response 11) Thank you, we modify the tables adding an additional column as you correctly suggest; we explained in the table legend the number of references referring to the papers which report the association described and we explained N° as number of papers dealing with diagnostic or prognostic association / N° of total analyzed papers dealing with the mentioned biomarker.

References:

12) The reference to PRISMA guidelines (line 130; ref 17) needs correcting.

Response 12) The correct reference is reported: Page, M.J.; McKenzie, J.E.; Bossuyt, P.M.; Boutron, I.; Hoffmann, T.C.; Mulrow, C.D.; Shamseer, L.; Tetzlaff, J.M.; Akl, E.A.;  Brennan, S.E.; et al. The PRISMA 2020 Statement: An Updated Guideline for Reporting Systematic Reviews. BMJ 2021, 372, n71. PMID: 33782057

13) There are various places in the text with numbers in brackets (e.g 32324430 in line 109) which presumably are references. Lines 290, 293, 294 likewise.

Response 13) Yes, thank you, we corrected.

Minor points:

14) Line 50: it is unclear what exactly you mean by anamnestic in this context.

Response 14) We provided to explain what anamnestic data concern.

15) Line 165: data extraction section: “disagreements on search strategy”; surely that should have been resolved before carrying out the search?

Response 15) Yes, we provided to modify this sentence. The disagreement on search strategy was resolved before carrying out the search; only in the case of indecision of some papers this is resolved by one author.

16) Line 287: tongue base rather than tongues.

Response 16) Sure, thank you.

We would like to thank the Editor and all the Reviewers again, for the valuable comments on the manuscript. We did our best to revise the manuscript following the suggestions, provide sufficient explanations, and perform a relevant revision on the manuscript. Moreover, we provided to add the number of PROSPERO of this systematic review.

Please let us know if there is any part we need to revise further. We really would like to express our deepest gratitude to the Editor and the Reviewers for their constructive comments helping us to improve our manuscript and giving us the opportunity for this revision.

Sincerely,

Daria Maria Filippini (MD, PhD student)

Reviewer 3 Report

The authors have provided a systematic review of diagnostic and prognostic biomarkers for head and neck cancer of unknown primary (HNCUP): an unmet clinical need.

Following guidelines in PRISMA protocol, authors searched the Web of Science (n=474) and PubMed (n=230) databases to retrieve the scientific literature relevant to HNCUP and with systematic screening reviewed 23 full-text research articles. The screening procedure itself reveals a small niche for HNCUP research and is also a challenging field to navigate diagnosis and prognosis. The general features of the chosen articles such as year, countries, and study types are well tabulated.

The authors nicely summarized the articles by providing a flow chart for systematic diagnosis of HNCUP. The chosen articles were distilled into a summary of different biological samples used for diagnosis, quantification methods, types, and a list of molecular biomarkers. The diagnostic and prognostic values of biomarkers further add insights into the significance of the biomarkers. The two main groups of biomarkers are related to HPV and EBV testing and are described in sufficient detail.

The authors propose miRNA expression profiling to determine the primary tumor of CUP and highlight the need for better characterization of the molecular profiling and the development of tissue-of-origin classifiers to improve the diagnosis, staging, and therapeutic management of patients with HNCUP.

Overall the systematic review provides a very useful summary of HNCUP studies published so far and a list of biomarkers for diagnosis and prognosis of HNCUP with a set of recommendations for clinical practitioners and researchers.

Author Response

THANK YOU VERY MUCH TO REVIEWER 3 FOR YOUR KIND RESPONSE AND VERY APPRECIATED YOUR INSIGHT IN THE REVIEW.

Daria Maria Filippini 

Round 2

Reviewer 2 Report

The paper reads much better with the incorporated changes.

I had previously picked up on your not having registered your protocol with PROSPERO but had not thought that its absence was a significant cause of bias so had not included this in my previous comments.

As you know, the purpose of protocol registration is to ensure that the review is carried out according to predefined criteria, and that the methods of analysis are not subsequently changed during the analysis stage to fit the data or the investigators' views on the topic. I see from the PROPERO entry (attached) that the study was first submitted to PROSPERO on March 4th and registered on March 15th. I first received your paper for review on March 9th, and at the bottom of this review proforma, the date submitted is given as March 4th, so the appearance is that all stages of the review were completed prior to PROSPERO registration. In your PROSPERO registration, you clearly state that at that point only the preliminary searches and piloting of the study selection process had been started and that no part of the review process had been completed. The final part of the PROSPERO registration states: The record owner confirms that the information they have supplied for this submission is accurate and complete and they understand that deliberate provision of inaccurate information or omission of data may be construed as scientific misconduct. 

There is clearly some inconsistency here for which I am sure the Editors will require an explanation. 
